# Phosphonate Inhibitors of Pyruvate Dehydrogenase Perturb Homeostasis of Amino Acids and Protein Succinylation in the Brain

**DOI:** 10.3390/ijms232113186

**Published:** 2022-10-29

**Authors:** Artem V. Artiukhov, Vasily A. Aleshin, Irina S. Karlina, Alexey V. Kazantsev, Daria A. Sibiryakina, Alexander L. Ksenofontov, Nikolay V. Lukashev, Anastasia V. Graf, Victoria I. Bunik

**Affiliations:** 1Department of Biokinetics, A. N. Belozersky Institute of Physicochemical Biology, Lomonosov Moscow State University, 119234 Moscow, Russia; 2Department of Biochemistry, Sechenov University, 105043 Moscow, Russia; 3Department of Clinical Medicine, Sechenov University, 105043 Moscow, Russia; 4Faculty of Chemistry, Lomonosov Moscow State University, 119234 Moscow, Russia; 5Faculty of Biology, Lomonosov Moscow State University, 119234 Moscow, Russia; 6Faculty of Bioengineering and Bioinformatics, Lomonosov Moscow State University, 119234 Moscow, Russia

**Keywords:** branched chain 2-oxo acids, 2-oxoglutarate dehydrogenase, pyruvate dehydrogenase, phosphonate and phosphinate analogs of pyruvate, protein succinylation, protein acetylation, protein glutarylation, sirtuin 3, sirtuin 5, anxiety

## Abstract

Mitochondrial pyruvate dehydrogenase complex (PDHC) is essential for brain glucose and neurotransmitter metabolism, which is dysregulated in many pathologies. Using specific inhibitors of PDHC in vivo, we determine biochemical and physiological responses to PDHC dysfunction. Dose dependence of the responses to membrane-permeable dimethyl acetylphosphonate (AcPMe_2_) is non-monotonous. Primary decreases in glutathione and its redox potential, methionine, and ethanolamine are alleviated with increasing PDHC inhibition, the alleviation accompanied by physiological changes. A comparison of 39 brain biochemical parameters after administration of four phosphinate and phosphonate analogs of pyruvate at a fixed dose of 0.1 mmol/kg reveals no primary, but secondary changes, such as activation of 2-oxoglutarate dehydrogenase complex (OGDHC) and decreased levels of glutamate, isoleucine and leucine. The accompanying decreases in freezing time are most pronounced after administration of methyl acetylphosphinate and dimethyl acetylphosphonate. The PDHC inhibitors do not significantly change the levels of PDHA1 expression and phosphorylation, sirtuin 3 and total protein acetylation, but increase total protein succinylation and glutarylation, affecting sirtuin 5 expression. Thus, decreased production of the tricarboxylic acid cycle substrate acetyl-CoA by inhibited PDHC is compensated by increased degradation of amino acids through the activated OGDHC, increasing total protein succinylation/glutarylation. Simultaneously, parasympathetic activity and anxiety indicators decrease.

## 1. Introduction

The thiamine diphosphate (ThDP)-dependent pyruvate dehydrogenase (PDH, 1.2.4.1) functions within the multienzyme pyruvate dehydrogenase complex (PDHC) that links cytosolic glycolysis with the mitochondrial tricarboxylic acid (TCA) cycle. When oxidizing the glycolytic product pyruvate, PDHC produces the TCA cycle substrate acetyl-CoA and NADH, which is oxidized in the mitochondrial electron transport chain. In addition, the PDHC reactive acetyl intermediates and the product acetyl-CoA donate acetyl moieties for post-translational modifications of proteins, which are of regulatory significance [1,2,3]. Although there are different sources of acetyl-CoA for the protein acetylation, participation of PDHC in acetylation of histones is shown [4,5] along with the observations of PDHC [4] and its PDH component [5,6] in cell nuclei. PDHC-dependent acetylation of metabolic proteins has not received significant attention, apart from early studies on the autoacetylation of PDHC [1]. The goal of the current work is to reveal the consequences of inhibition of the PDHC reaction on the brain biochemical indicators of the major role of PDHC in mitochondrial metabolism, along with impact of PDHC inhibition on the physiological state of animals.

Different pharmacological tools are available for PDHC regulation. Thiamine (vitamin B1) and its pharmacological forms are precursors for a PDHC coenzyme, ThDP [7,8]. In addition to the thiamine-related activators of PDHC reaction, antagonists of thiamine [7,8,9] and of another coenzyme of PDHC, covalently bound lipoic acid [10,11,12], may be used to inhibit the reaction. However, these compounds would affect all the complexes of dehydrogenases of 2-oxo acids. In contrast, inhibition of PDHC by synthetic analogues of pyruvate is an effective way to target specifically PDHC, not affecting the other family members [9,13,14]. Hence, in this work, we apply the phosphonate inhibitors of PDH to reveal the consequences of the brain PDHC dysfunction for the biochemical and physiological parameters. Using the membrane penetrating dimethyl ester of acetyl phosphonate (AcPMe_2_), we characterize the dose dependence of the changes. Furthermore, we use a fixed dose of the different phosphonate analogs of pyruvate, shown in Figure 1, to characterize their relative efficacy in vivo.

## 2. Results

### 2.1. Dose-Dependent Effects of Dimethyl Ester of Acetyl Phosphonate on Metabolism and Physiology

The fully esterified phosphonate AcPMe_2_ (Figure 1) is known to cause a strong viability decrease upon its incubation with cell cultures [14,15]. No charged groups allow AcPMe_2_ to diffuse through biological membranes, whereas charged analogs require protein transporters/channels to penetrate the membrane. As a result, compared to AcPMe_2_, intracellular concentration of the charged analogs is expected to be a more complex function of their extracellular concentration and intracellular transport. Hence, AcPMe_2_ has been chosen to study the concentration dependence of inhibition of the brain PDHC in vivo. The metabolic indicators of PDHC function in the TCA cycle, used to characterise the action of PDHC inhibitors, are shown in Figure 2. Apart from PDHC activity, important enzymatic indicators of the PDHC-dependent flux through the TCA cycle are employed. They are represented by the activities of (i) the TCA-cycle-limiting 2-oxoglutarate dehydrogenase complex (OGDHC), (ii) the enzymes metabolising malate, i.e., malate dehydrogenase (MDH) and decarboxylating malic enzyme (ME) important for anaplerosis, and (iii) glutamine synthetase (GS), whose function is linked to the amino acid degradation occurring in the TCA cycle [16]. Accordingly, the profile of the brain amino acids and related compounds is a useful indicator of the TCA cycle substrate fluxes [16,17].

In accordance with the reversibility of the phosphonate inhibitors binding to PDHC [9], no alterations in the activity of PDHC assayed upon a strong dilution of the brain homogenates in the reaction medium is observed (Figure 2). However, dose-dependent changes in many metabolic indicators (Figure 2A) point to strong perturbation of brain metabolism upon the PDHC inhibition in vivo.

At the low AcPMe_2_ dose (0.02 mmol/kg), significant decreases in glutathione and its redox potential, ethanolamine, methionine and valine are observed (Figure 2A,B). Thus, these changes are primary indicators of PDHC impairment. Upon the administration of a high AcPMe_2_ dosage (0.1 mmol/kg), decreases in these primary indicators, except for that of valine, are alleviated (Figure 2B). The switch is accompanied by changes in the activities of OGDHC, MDH and GS (Figure 2C). Remarkably, activities of PDHC and the pyruvate-producing malic enzyme (ME), along with a known indicator of the pyruvate oxidation in the TCA cycle, alanine, do not significantly change. The finding suggests homeostatic stabilization of the pyruvate/alanine ratio at the metabolite level through the network of transamination reactions. In contrast, the activity of malate dehydrogenase (MDH) producing oxalacetate for condensation with acetyl-CoA, is decreased (Figure 2C). The decrease is in accord with an inhibited acetyl-CoA flux into the TCA cycle. At the same time, the activity of OGDHC increases in response to the high level of PDHC inhibition in vivo, along with a decrease in the brain level of glutamate. Not only does the activated OGDHC promote glutamate oxidation in the TCA cycle, but also the glutamate usage for biosynthesis obviously contributes to the decrease in glutamate levels. Indeed, at a high AcPMe_2_ dose, the glutathione level returns to level of the control one (Figure 2B), and production of glutamine from glutamate by glutamine synthetase (GS) is activated (Figure 2C). The increased activity of GS indicates increased ammoniac production due to the degradation of amino acids in the TCA cycle with activated OGDHC. Indeed, at the strong PDHC inhibition in vivo, the branched-chain amino acids leucine and isoleucine decrease (Figure 2C), in addition to the primarily decreased valine (Figure 2B).

The high dose of AcPMe_2_ also affects the physiological state; the root mean square of the successive differences in RR-intervals (RMSSD) of an electrocardiogram (ECG) decrease along with an increase in the locomotor activity, estimated by crossing the lines in the Open Field Test (Figure 3). The biphasic response of physiological parameters to increasing doses of the PDHC inhibitor is evident from the significant differences between the effects of the two different doses of AcPMe_2_, although the low dose shows no effect vs. control animals. This is manifested in the line crossing and freezing time (Figure 3), pointing to the opposite changes in these parameters at the two doses of the inhibitor. Thus, the physiological effects of PDHC inhibition shows a complex dependence on the inhibitor dosage, similar to the brain biochemical parameters.

### 2.2. Comparison of the Biochemical and Physiological Effects of the Phosphinate and Phosphonate Inhibitors of PDHC

The biochemical and physiological effects of the phosphinate (AcMeP) and phosphonate (AcP, AcPMe, AcPMe_2_) analogs of pyruvate (Figure 1), known to be of different inhibition power from in vitro studies [9,14,18], have been compared after administration to animals of the same dosage of the drugs (0.1 mmol/kg). In vitro, the inhibitory power of the charged analogs decreases in the order AcMeP < AcP < AcPMe, but inside the cells, cellular esterases may de-esterify both the mono- (AcPMe) and dimethyl (AcPMe_2_) esters to AcP. These considerations are in accord with the automatically obtained clusters of the inhibitors-induced changes in the brain biochemical parameters, shown in Figure 4A. That is, the clusters shown at the left of Figure 4A, indicate that the changes induced by the strongest in vitro phosphinate inhibitor AcMeP are separated from the changes induced by the phosphonate inhibitors AcP, AcPMe and AcPMe_2_, with the latter three forming a common cluster.

Furthermore, none of the inhibitors significantly affect the assayed levels of PDHC activity and alanine, but all the inhibitors increased the activity of OGDHC, decreasing the levels of glutamate and branched-chain amino acids, with NAD^+^ mostly decreased as well (Figure 4B). Although MDH is decreased by AcPMe_2_ only, the primary changes, occurring at the low dose of AcPMe_2_, i.e., the decreases in the level and redox potential of glutathione or in the levels of ethanolamine and methionine (Figure 2B), are not observed for any of the analogs (Figure 4C). As a result, the most essential changes observed after administration of the fixed dose of all the analogs (Figure 4B,C), are similar to those after administration of the high dose of AcPMe_2_ (Figure 2). Thus, administration of all the employed analogs results in a PDHC inhibition level that corresponds to the second phase of the induced changes.

The high inhibition level by all the analogs is further supported by physiological changes. The treatment factor is of significance for RMSSD of ECG (*p* = 0.041) and duration of freezing (*p* = 0.012). A decrease after administration of PDHC inhibition, with the most significant decreases being vs. control animals, was observed after administration of the strongest in vitro inhibitor AcMeP and/or membrane-penetrating AcPMe_2_ (Figure 5).

### 2.3. Action of the PDHC Inhibitors on the PDHA Expression and Its Active Site Phosphorylation at Ser293

In view of the regulation of the PDHC activity by phosphorylation of its rate-determining component at the active site residue Ser293 of PDHA1 subunit, we assessed the action of the PDHC inhibitors on enzyme expression and phosphorylation. After the administration of the inhibitors at 0.1 mmol/kg, no significant changes are observed in either the PDHA1 expression (Figure 6A) or its phosphorylation at Ser293 (Figure 6B). Moreover, phosphorylation normalized to the expression level does not change either (Figure 6C).

### 2.4. Changes in Total Levels of the Brain Protein Acylations Induced by the PDHC Inhibitors

Changes in acylations of the brain proteins and expression of deacylases sirtuins 3 and 5 upon inhibition of the acetyl-CoA production by PDHC are shown in Figure 7. The levels of total protein acetylation and mitochondrial deacetylase sirtuin 3 did not significantly change (Figure 7A,B). However, the PDHC inhibition induces significant changes in the system of the negatively charged acylations; levels of total succinylation and glutarylation of the brain proteins are increased, while expression of the corresponding deacylase sirtuin 5 is down-regulated (Figure 7C–E).

Correlation analysis of the brain components of the acylation system in the pooled sample of studied animals (Figure 8) points to significant negative correlations of the levels of total succinylation with NAD^+^ and of the levels of total glutarylation with sirtuin 3. Furthermore, sirtuin 3 is positively correlated with total acetylation of the brain proteins, whereas sirtuin 5 is negatively correlated with the OGDHC activity and positively with NAD^+^ levels. Remarkably, the phosphorylation of the PDHA1 subunit of PDHC positively correlates with the level of total succinylation of the brain proteins. This correlation complements the finding that PDHC inhibition increases total succinylation (Figure 7C), as PDHA1 phosphorylation also decreases PDHC activity. Thus, across the studied conditions, the brain protein succinylation increases with increasing inactivation of PDHC by its phosphorylation or inhibition and decreasing the desuccinylase substrate NAD^+^. In contrast, total acetylation of the brain proteins has an intrinsic adjustment mechanism, increasing expression of sirtuin 3 along with increased protein acetylation. Finally, our data show different regulatory networks for succinylation and glutarylation of the brain proteins. On one hand, levels of both types of the acylation increase upon the PDHC inhibition simultaneously with down-regulation in sirtuin 5 (Figure 7) and up-regulation of OGDHC activity (Figure 4), which are negatively correlated (Figure 8). On the other hand, the correlation analysis indicates that under the same set of conditions, succinylation and glutarylation show different interdependences on the factors essential for the acylation levels (Figure 8). Succinylation is strongly linked to the availability of the desuccinylase substrate NAD^+^, whose decrease expectedly increases succinylation. In contrast, glutarylation shows a more complex control, being strongly linked to the acetylation system. That is, glutarylation decreases along with increasing expression of the deacetylase sirtuin 3.

## 3. Discussion

The dose dependence of the brain biochemical response to PDHC inhibition, shown in this study, demonstrates biphasic changes in the brain metabolism with increasing PDHC inhibition. In view of the recently shown biphasic response of the brain metabolism to inhibition of OGDHC [19], we conclude that biphasicity is an essential feature of the perturbation in these critical branch points of animal metabolism, which are interconnected through the TCA cycle. In both cases, the primary changes in the levels of critical metabolites induce a homeostatic response that is directed to alleviation of such changes. However, this process brings about secondary changes in other parts of the perturbed metabolic network, affecting physiological indicators of anxiety and parasympathetic activity.

The primary changes upon the PDHC inhibition (Figure 2B) affect the metabolites essential for the glutathione redox-state (glutathione), phospholipid biosynthesis (ethanolamine) [20,21], or anabolic reactions requiring adenosyl methionine (methionine). Remarkably, dihydrolipoic acid, an intermediate of the PDHC reaction, reduces peroxidized phospholipids, including oxygenated phosphatidylethanolamine [22]. One of the most abundant mammalian phospholipids, phosphatidylethanolamine, is known to participate in autophagy, ferroptosis, endoplasmic reticulum stress and Parkinson’s disease [21,22]. Phosphatidylethanolamine alleviates mitochondrial dysfunction induced by insufficiency of the mitochondria-specific phospholipid cardiolipin [23] that activates PDHC [24]. Translocation of cardiolipin from the inner to outer mitochondrial membrane, known to induce mitophagy [25], may thus bring about deactivation of the cardiolipin-activated PDHC, similar to the PDHC activity loss due to inhibition. In view of these independent data on functional interactions between PDHC, the phospholipids and mitochondrial dysfunction, our finding that the PDHC inhibition in the brain causes biphasic changes in the brain level of ethanolamine (Figure 2) indicates increased synthesis of phosphatidylethanolamine from ethanolamine as a primary response to PDHC inhibition. Remarkably, biosynthesis of phosphatidylethanolamine occurs from either ethanolamine or serine [23]. The secondary decreases in the brain serine levels are a universal consequence of the action of all PDHC inhibitors at a high dose (Figure 4A). The finding suggests an increase in the phosphatidylethanolamine formation from serine, compensating for the primary decrease in ethanolamine content at the low dose of PDHC inhibition (Figure 2B). The inhibition-induced switch to another biosynthetic pathway may contribute to the normalization of ethanolamine levels observed after the primary decrease (Figure 2B).

By decreasing the acetyl-CoA entry to the TCA cycle, PDHC inhibition stimulates the entry of amino acids to the cycle through activation of the TCA-cycle-limiting OGDHC. The increased degradation of amino acids upon PDHC inhibition is manifested in increased activity of the ammonia-assimilating glutamine synthetase and decreased levels of glutamate and branched chain amino acids (Figure 2). Thus, the brain energy demands upon PDHC inhibition are addressed by usage of amino acids as alternative energy substrates. Obviously, activation of glycolysis cannot substitute for the impairment in energy production by the brain mitochondria with inhibited PDHC. Moreover, downregulation of malate dehydrogenase activity with increasing the PDHC inhibition manifests decreased entry of reducing equivalents from cytosol to mitochondria through the malate-aspartate shuttle, not excluding downregulation of the pyruvate production by glycolysis upon PDHC inhibition.

It is worth noting that the metabolic, and particularly energetic, impact of the OGDHC upregulation, as well as the upregulation itself, are strongly conditional, depending on the physiological state, damage level and cells/tissues under consideration [26,27]. Functional upregulation of the complex may support no decreases in the neuronal levels of NADP(H) or ATP up to a certain level of the OGDHC inhibition by its specific inhibitor succinyl phosphonate [26,28]. However, administration of a high ethanol dose may also induce OGDHC upregulation, yet neuronal levels of both NAD(P)H and ATP significantly decrease in the medium with a high ethanol content [26]. Furthermore, both ethanol and succinyl phosphonate may upregulate OGDHC, but their combination does not provide an additive effect, decreasing the OGDHC upregulation instead. Thus, upregulation is impaired upon increasing damage, which may explain promotion of the ethanol-induced energetic impairment by succinyl phosphonate in cultured neurons [26]. It should also be noted in this regard that an enzyme upregulation determined in homogenates under the standard assay conditions employing saturating concentrations of all the substrates and coenzymes, estimates the active enzyme quantity rather than the real substrate flux through the enzyme. The flux is determined not only by the level of the active enzyme, but also by the levels of its substrates/coenzymes. For instance, one of the mechanisms, effected in the brain to compensate for the decreased flux through the succinyl-phosphonate-inhibited OGDHC, is an increase in the brain levels of the OGDHC coenzyme ThDP [29]. Dependent on conditions, increases in the active enzyme quantity may thus occur by different means, not always detectable in the standard assay conditions. Overall, prediction of the metabolic impact of the changes in a single enzyme activity assayed under standard conditions requires consideration of the enzymatic network and the levels of the system damage/reparation.

As discussed above, complementary regulation of activities of PDHC and OGDHC is found in the current study on the PDHC inhibition in the male brain cortex, where PDHC inhibition elevates OGDHC function. Similarly, in the previous study on the OGDHC inhibition, the changes in the PDHC activity mirror those in the OGDHC activity [19]. The complementary regulation of the PDHC and OGDHC activities in the brain cortex of male rats is manifested in significant negative correlation between the levels of these activities in the homogeneous sample of control male rats (*n* = 24) [19]. The negative correlation suggests concerted variations in the functional expression of PDHC and OGDHC, most probably corresponding to interindividual physiological differences in the brain TCA cycle fluxes. During the day, however, disparate changes in OGDHC and PDHC activities are observed in the brain cortex of male rats, accompanied by the disappearance of the negative correlation between the OGDHC and PDHC activities [30]. Thus, concerted regulation of the two multienzyme complexes may be uncoupled to support diurnal rhythms. As a result, different correlations between the PDHC and OGDHC functions may be observed in different animal samples and/or upon varied treatments. Indeed, in the female rat cerebellum under physiological conditions, the levels of the metabolic indicators of the PDHC and OGDHC function, i.e., alanine and glutamate, correspondingly, do not strongly correlate to each other or to the OGDHC activity. However, short-term exposure of these rats to hypoxia induces a strong positive correlation between the levels of alanine and glutamate, and strong negative correlations of each indicator with the OGDHC activity [17,31]. Along with the changed correlations, simultaneous increases in the brain levels of alanine and glutamate after acute hypoxia indicate concerted changes in the substrate fluxes through the two multienzyme complexes, observed in the female rat cerebellum in response to the insult.

Remarkably, both OGDHC and PDHC inhibition induce increases in OGDHC activity and biphasic changes in glutathione metabolism. However, when OGDHC is inhibited, the OGDHC activity increase is a primary response, followed by the activity decrease. Accordingly, administration of both the OGDHC and PDHC inhibitors affects brain glutathione metabolism, yet the primary effect of the OGDHC inhibition is manifested in the glutathione disulfide levels [19], whereas that of the PDHC inhibition is seen in the levels of reduced glutathione (Figure 2).

A compensatory response to PDHC inhibition could include the dephosphorylation of the first component of the complex to increase PDHC activity. However, this does not occur, at least under the induced levels of the enzyme inhibition (Figure 6). The total acetylation of the brain proteins does not change either. In contrast, succinylation and glutarylation, dependent on the function of OGDHC or its second and third components, which are common for OGDHC and the analogous 2-oxoadipate dehydrogenase complex (OADHC) do (Figure 7A). Thus, the total acylation of the brain proteins is in accord with the changed activity of OGDHC and no changes in that of PDHC (Figure 2 and Figure 4). Nevertheless, the level of PDHA1 phosphorylation shows a strong positive correlation with total succinylation (Figure 8), revealing the interconnected regulation of the two post-translational modifications of the brain proteins. The lower the PDHC activity because of PDHA1 phosphorylation, the higher the protein succinylation level. This result of the correlation analysis is supported by the same action on the succinylation of the PDHC inhibitors (Figure 7C) and PDHC inactivation by phosphorylation (Figure 8). This relationship most likely manifests in the competition between the parallel reactions of succinylation and acetylation of the protein lysine residues. Moreover, total succinylation increases along with the increase of OGDHC activity (Figure 4B), which is also negatively correlated with the desuccinylase sirtuin 5 (Figure 8). Finally, total succinylation correlates negatively with the levels of the deacylases substrate NAD^+^. As a result, under the studied conditions of the PDHC inhibition, the levels of the OGDHC activity, PDHA1 phosphorylation and NAD^+^ contribute the most to protein succinylation.

Reduced acetyl-CoA production by PDHC is known to decrease biosynthesis of acetylcholine in the brain or neurons [32,33]. This is observed in the thiamine deficiency states [34,35,36,37,38], exposures to β-amyloid [39], aluminium, zinc ions or nitric oxide [37,38,40]. The observed decreases in an indicator of parasympathetic activity RMSSD and in duration of freezing (Figure 3 and Figure 5) are in accordance with the PDHC-inhibition-induced decrease in acetylcholine, shown in the independent studies mentioned above.

Thus, the characterized metabolic action of specific inhibitors of PDHC unravels molecular mechanisms involved in the brain homeostatic response to perturbed PDHC function, the secondary metabolic perturbations and their physiological impact.

## 4. Materials and Methods

### 4.1. Reagents

Sodium salt of AcPH was synthesized as previously described [14]. Sodium salt of AcMeP was synthesized via different procedures according to [18,41]. Methyldichlorophosphine (0.1 mol, 11.69 g, 8.84 mL) was added to trimethyl orthoacetate (0.228 mol, 27.35 g, 29 mL) at −30 °C. The resulting mixture was stirred at an ambient temperature for 18 h. The excess of trimethyl orthoacetate was removed under reduced pressure. The product (methyl (1,1-dimethoxyethyl)methylphosphinate) was isolated by vacuum distillation. Yield: 15.1 g (83%), b.p. 81–83 °C/0.9 mm. NMR ^1^H (CDCl_3_), δ, ppm: 3.76 (d, *J* = 10.2 Hz, 3H, (CH_3_O)P(O)), 3.37 (d, *J* = 6.6 Hz, 6H, C(OCH_3_)_2_), 1.46 (d, *J* = 2.3 Hz, 3H, CH_3_C), 1.42 (s, 3H, CH_3_P). NMR ^13^C (CDCl_3_), δ, ppm: 100.7 (d, *J* = 143.3 Hz, CCH_3_), 51.7 (d, *J* = 6.7 Hz, (CH_3_O)P(O)), 49.8 (d, *J* = 5.9 Hz, C(OCH_3_)_2_), 49.6 (d, *J* = 6.7 Hz, C(OCH_3_)_2_), 19.0 (d, *J* = 12.7 Hz, CCH_3_), 10.7 (d, *J* = 89.4 Hz, CH_3_P). NMR ^31^P (CDCl_3_), δ, ppm: 49.5.

To obtain sodium (1,1-dimethoxyethyl)methylphosphinate, a solution of methyl (1,1-dimethoxyethyl)methylphosphinate (10 mmol, 1.82 g) and NaI (11 mmol, 1.65 mg) in 11.4 mL of dry methylethylketone was refluxed for 2 h under argon. The resulting precipitate was filtered off, washed twice with 5 mL of dry acetone and dried in vacuo. Yield: 1.56 g (82.2%), m.p. 186–188 °C. NMR ^1^H (D_2_O), δ, ppm: 3.36 (s, 6H, C(OCH_3_)_2_), 1.43 (d, *J* = 10.1 Hz, 3H, CH_3_C), 1.27 (d, *J* = 13.4 Hz, 3H, CH_3_P). NMR ^13^C (D_2_O), δ, ppm: 101.4 (d, *J* = 141.6 Hz, CCH_3_), 49.8 (d, *J* = 5.9 Hz, C(OCH_3_)_2_) 18.4 (d, *J* = 11.0 Hz, CCH_3_), 13.3 (d, *J* = 89.4 Hz, CH_3_P). NMR ^31^P (D_2_O), δ, ppm: 37.5.

Sodium (1,1-dimethoxyethyl)methylphosphinate (5.9 mmol, 1.12 g) was further dissolved in a mixture of 5.9 mL of glacial acetic acid and 0.35 mL of water. The solution was stirred at ambient temperature for 24 h. The solvent was evaporated to dryness and residue was triturated with 30 mL of acetone and filtered off to give sodium AcMeP as a white solid. Yield: 0.8 g (94.3%), m.p. 195–197 °C. NMR ^1^H (D_2_O), δ, ppm: 2.42 (d, *J* = 3.8 Hz, 3H, CH_3_C(O)), 1.36 (d, *J* = 14.2 Hz, 3H, CH_3_P). NMR ^13^C (D_2_O), δ, ppm: 224.5 (d, *J* = 109.6 Hz, C(O)P(O)), 27.9 (d, *J* = 42.2 Hz, CH_3_C(O)), 12.3 (d, *J* = 95.3 Hz, CH_3_P). NMR ^31^P (D_2_O), δ, ppm: 27.1.

AcPMe_2_ was synthesized from dimethyl phosphite and acetyl chloride as described previously [14]. Partial hydrolysis of AcPMe_2_ with NaI according to [14] resulted in the sodium salt of AcPMe. Complete hydrolysis with NaHCO_3_ resulting in disodium salt of AcP was done according to [42]. Bromotrimethylsylane (40 mmol, 6.12 g, 5.3 mL) was added dropwise to AcPMe_2_ (10 mmol, 1.52 g) and stirred at 0 °C under argon. The resultant solution was stirred at an ambient temperature overnight. The excess of bromotrimethylsilane was removed under reduced pressure. Aqueous NaHCO_3_ (1M, 20 mL) was added at 0 °C and the solution stirred at 0 °C for 1 h. Water was evaporated to dryness under reduced pressure and the residue was washed with 10 mL of absolute ethanol to yield a solid white product. Yield: 1.56 g (93%). NMR ^1^H (D_2_O), δ, ppm: 2.34 d (3H, CH_3_C(O), *J* = 3.6 Hz). NMR ^13^C (D_2_O), δ, ppm: 228.2 (d, *J* = 157.0 Hz, C(O)P(O)), 29.8 (d, *J* = 43.7 Hz, CH_3_C(O)). NMR ^31^P (D_2_O), δ, ppm: −0.1.

NAD^+^ was obtained from Gerbu (Heidelberg, Germany), and oxidized glutathionefrom Calbiochem (La Jolla, CA, USA). Formate dehydrogenase for NAD^+^ assays was obtained from the Federal Research Center of Biotechnology/Innotech MSU (Moscow, Russia). All other reagents were of the highest purity available and obtained from Sigma-Aldrich (Helicon, Moscow, Russia).

### 4.2. Animal Experiments

Animal experiments were performed according to the Guide for the Care and Use of Laboratory Animals published by the European Union Directives 86/609/EEC and 2010/63/EU and were approved by the Bioethics Committee of Lomonosov Moscow State University (protocols number 69-o from 09.06.2016 and 139-a from 11.11.2021). Wistar male rats were kept in standard conditions with 12 h light and 12 h dark day cycle, with free access to water and food. Phosphinate and phosphonate analogues of pyruvate were dissolved in deionized water to obtain 0.2 M or 1 M solutions and were administered intranasally at 0.02 mmol/kg (AcPMe_2,_ only) or 0.1 mmol/kg dosage (all phosphonates), respectively. A physiological solution (0.9% NaCl) was administered to the control animal group. Then, 24 h after the administration, the rats were subjected to physiological tests and sacrificed by decapitation as described before [43,44]. Immediately after the decapitation, the animal brain was excised and the brain cortex was separated on ice and frozen in liquid nitrogen within 90 s after decapitation. The tissue samples were stored at −70 °C until biochemical assays could be performed.

### 4.3. Physiological Tests

Estimation of behavioral activity was performed using the “Open Field” test as described previously [43,45]. ECG was recorded for 3 min using a non-invasive electrode placement [43]. The heart rate variability parameters calculated from ECG recordings are described previously [43].

### 4.4. Metabolite and Enzyme Activity Assays

The quantification of low-molecular weight metabolites was done in methanol/acetic acid extracts of rat cerebral cortex and prepared according to previously published research [46]. Amino acids and other amino compounds were quantified using cation-exchange chromatography with post-column ninhydrine derivatization [46]. Quantitave determination of other metabolites, including NAD^+^ and oxidized glutathione, was performed according to published protocols as described elsewhere [19]. The levels of all metabolites were calculated in µmoles per gram of tissue fresh weight (gFW).

Enzyme activities were measured in cerebral cortex homogenates prepared as described previously [16,19,47]. Spectrophotometric assays for glutamate dehydrogenase, malate dehydrogenase, NADP^+^-dependent malic enzyme, glutamine synthetase, PDHC and OGDHC are described in previous papers [16,19,47]. For the OGDHC assay, the blank media did not contain coenzyme A. NADP^+^-dependent isocitrate dehydrogenase was assayed in a previously described medium [47]. The reaction was started with addition of 3 µL of homogenate and monitored for 10 min. The resulting reaction slope was corrected for blank reaction assayed without D,L-isocitrate. The assayed enzyme activities were expressed as µmoles of substrate consumed or product generated per minute gFW.

### 4.5. Western Blotting

The tissue homogenates were diluted in Laemmli buffer and subjected to SDS-PAGE as described previously [44]. The levels of total protein succinylation and glutarylation were estimated by Western blotting using primary antibodies #PTM-401 and #PTM-1151, respectively, from PTM Biolabs (Chicago, IL, USA). The levels of total protein acetylation, sirtuin 3, sirtuin 5, PDHA1 and phosphorylated PDHA1 were estimated using antibodies #9841, #5490, #8782, #3205 and #37115, respectively, from Cell Signaling Technology (Danvers, MA, USA). All primary antibodies were used in a 1:2000 dilution. Secondary HRP-conjugated anti-rabbit antibody from Cell Signaling Technology, #7074 (Danvers, MA, USA) was used in 1:3000 dilution. Relative chemiluminescence was detected using ChemiDoc MP Imager (Bio-Rad, Hercules, CA, USA) and processed in Image Lab software v. 6.0.1 (Bio-Rad, Hercules, CA, USA). Normalization was done per total protein level in gel lane, determined via 2,2,2-tricholoroethanol staining as described elsewhere [48]. When comparing several membranes, the band intensities from different membranes were further normalized to the levels in common samples present in all the membranes. Raw images of membranes with visualized protein bands are presented in Appendix A.

### 4.6. Statistics and Data Analysis

Data are presented as mean ± standard error of the mean for each experimental group. Due to noticeable differences in the variances of the parameters in the experimental groups, the latter were compared using one-way Welch ANOVA with the recommended Games-Howell’s post hoc test, employed in GraphPad Prism 9.0 (GraphPad Software Inc., La Jolla, CA, USA) or userfriendlyscience package in R. The differences with *p* ≤ 0.05 were considered significant. Outliers (indicated as hollow points in graphs) were identified according to the iterative Grubb’s test (alpha = 0.01) in each experimental group and were excluded from statistical analysis. The number of animals in experimental groups (n) is indicated including outliers.

According to the Shapiro–Wilk test, some of the biochemical parameters were not normally distributed. Thus, for analysis of the components of protein acylation system, Spearman’s correlation coefficients were calculated for each pair of the parameters. Heatmaps including correlation matrices were prepared using RStudio 1.4 (Rstudio, PBC, Boston, MA, USA) and Adobe Illustrator 24.2 (Adobe, Inc., San Jose, CA, USA). The correlations with *p* ≤ 0.05 were considered significant. Where present, the row and column clustering of heatmaps was done using Euclidean distance and the “ward.D” agglomeration method employed in the pheatmap package in R.

## 5. Conclusions

In the current work, metabolic and physiological consequences of a short-term inhibition of PDHC in the brain cortex of male rats are characterized. First of all, PDHC inhibition affects the glutathione redox state (decreased glutathione), phospholipid biosynthesis (decreased ethanolamine) and anabolic reactions dependent on adenosyl methionine (decreased methionine). These changes are attenuated with increasing PDHC inhibition, which induces upregulation of the activity of the TCA-cycle-limiting OGDHC and downregulation of the malate dehydrogenase activity. As a result, degradation of amino acids in the TCA cycle is promoted to support the energy demands. Furthermore, succinylation/glutarylation of the brain proteins increases. PDHA phosphorylation does not show a strong response to the observed perturbations. The metabolic rearrangement upon increasing the PDHC inhibition is associated with decreased anxiety and reduced parasympathetic activity.

## Figures and Tables

**Figure 1 ijms-23-13186-f001:**
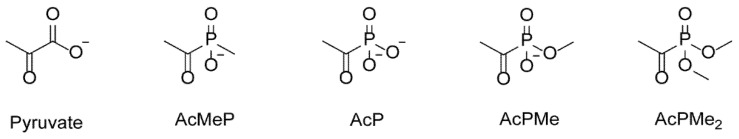
Structures of pyruvate and its phosphinate and phosphonate analogs, used in this study. Phosphinate analogues of pyruvate include acetyl (methyl) phosphinate (AcMeP). Phosphonate analogues of pyruvate include acetyl phosphonate (AcP) and its methyl (AcPMe) and dimethyl esters (AcPMe_2_).

**Figure 2 ijms-23-13186-f002:**
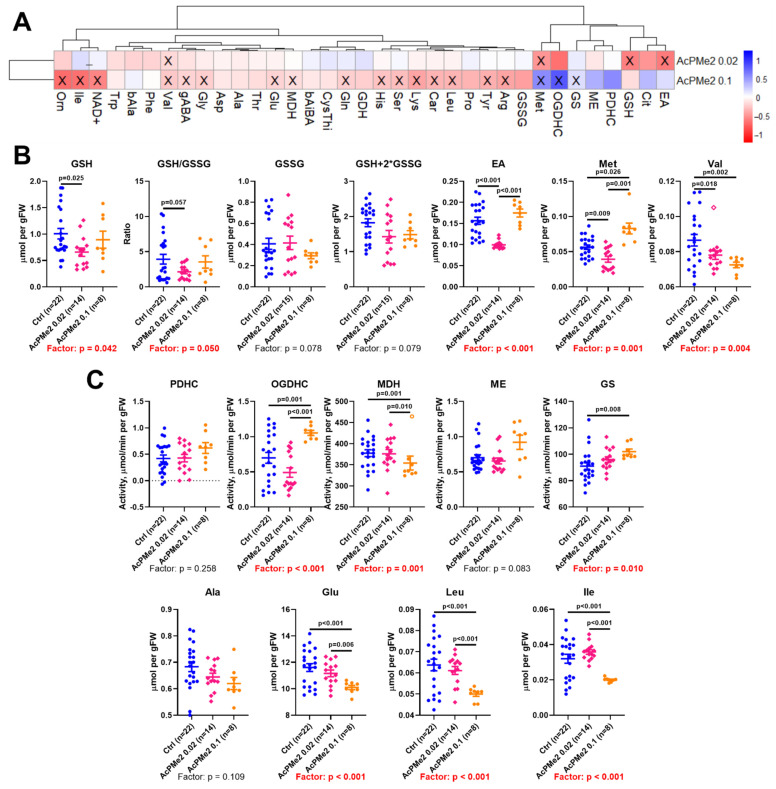
Dose-dependent effects of dimethyl ester of acetyl phosphonate (AcPMe_2_) on biochemical parameters of the rat cerebral cortex. (**A**) AcPMe_2_-induced changes in the levels of metabolites and enzymatic activities are presented as log2 of the fold change from non-treated animals. (**B**) Significant changes in the levels of amino acids and related compounds upon treatment with low dosage (0.02 mmol/kg) of AcPMe_2_. (**C**) Significant changes in the enzyme activities and levels of amino acids upon treatment with a high dosage (0.1 mmol/kg) of AcPMe_2_. Enzyme activities are presented as µmol/min per gram of tissue fresh weight (gFW), levels of metabolites–as µmol per gFW. Significant (*p* ≤ 0.05) differences between experimental groups, estimated by ANOVA with Tukey’s post-hoc test, are indicated as crossed cells on the heatmap (**A**; *p* ≤ 0.05 vs. Ctrl) or as exact *p*-values on the graphs (**B**,**C**). The ANOVA *p*-values for the treatment factor significance are shown below the graphs, with significant values of *p* ≤ 0.05 marked in red. Empty symbols denote outliers determined by the iterative Grubb’s test. The number of animals in the groups and dosages of the administered AcPMe_2_ are indicated on the X axes. Proteinogenic amino acids are abbreviated according to the standard three-letter code. Other abbreviations used are: Cntr—control, bAiBA—β-aminoisobutyrate, bAla—β-alanine, Car—carnosine, CysThi—cystathionine, EA—ethanolamine, gABA—γ-aminobutyrate, GSH—reduced glutathione, GSSG—oxidized glutathione, Orn—ornithine, GDH—glutamate dehydrogenase, MDH—malate dehydrogenase, ME—NADP^+^-dependent malic enzyme and GS—glutamine synthetase.

**Figure 3 ijms-23-13186-f003:**
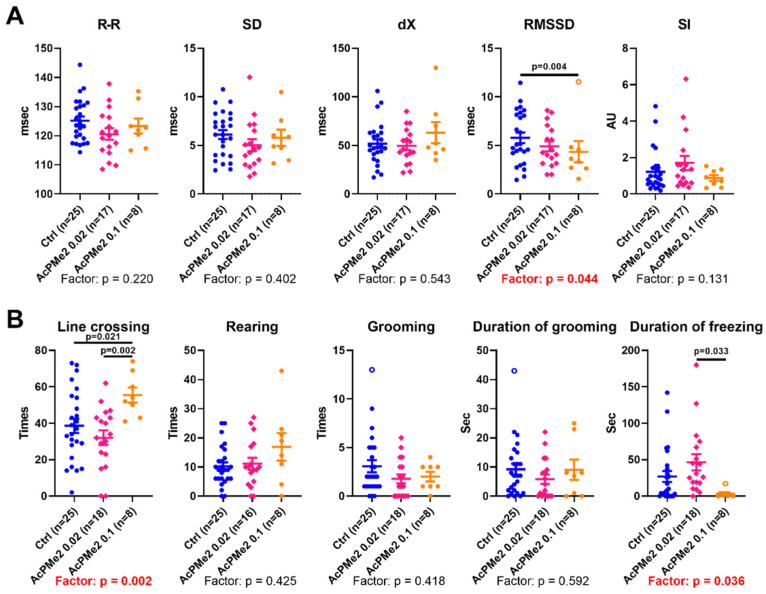
Changes in animal physiology after administration of the indicated dosages of AcPMe_2_. (**A**) Heart variability parameters obtained from ECG. (**B**) Behavioural parameters obtained in the Open Field Test. Significant differences of the treatments and between experimental groups are estimated and shown in the same way as in Figure 2.

**Figure 4 ijms-23-13186-f004:**
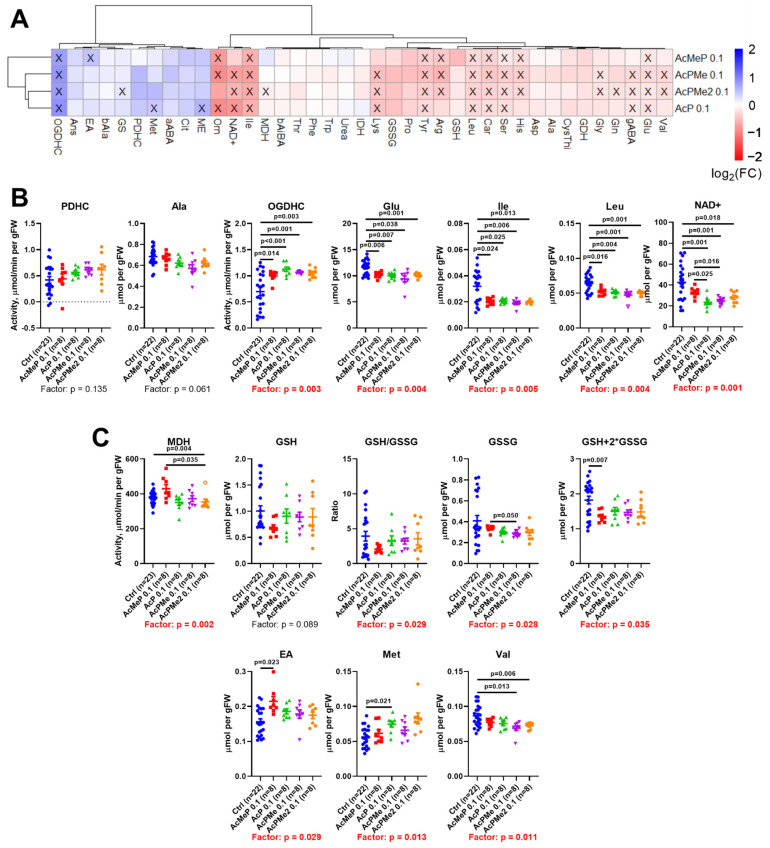
Comparison of the actions of the phosphinate and phosphonate analogues of pyruvate on the biochemical parameters of the rat cerebral cortex. (**A**) Levels of metabolites and enzymatic activities are presented as log2 of the fold change in the treated vs. untreated animals. (**B**) Activities of PDHC and associated enzymes. (**C**) Changes in MDH and primary indicators of PDHC function. Significant differences between experimental groups are estimated and shown in the same way as in Figure 2. Abbreviations are the same as in Figure 2. Other abbreviations: aABA—α-aminobutyrate, Ans—anserine, IDH—NADP^+^-dependent isocitrate dehydrogenase.

**Figure 5 ijms-23-13186-f005:**
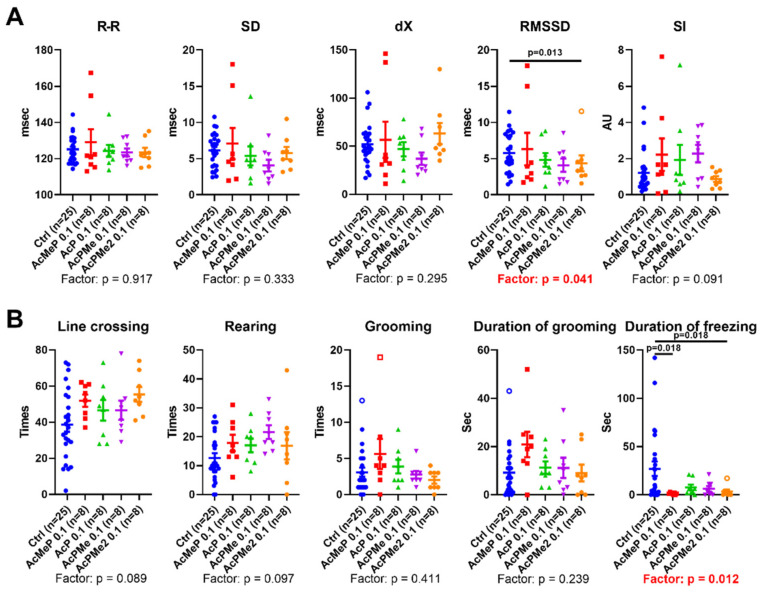
Effects of phosphonate and phosphinate analogues of pyruvate on animal physiology. (**A**) Heart variability parameters obtained from ECG. (**B**) Behavioural parameters obtained in the Open Field Test. Significant differences of the treatments and between experimental groups are estimated and shown in the same way as in Figure 2.

**Figure 6 ijms-23-13186-f006:**
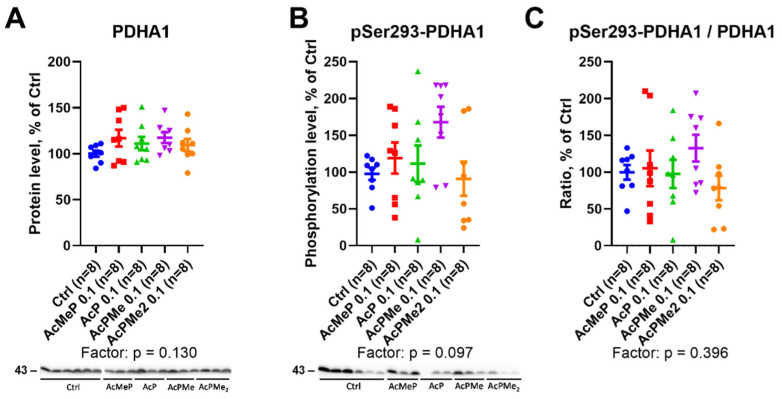
Expression and phosphorylation of PDHA1 subunit after the action of the PDHC inhibitors administered at 0.1 mmol/kg. (**A**) Protein levels of PDHA1. (**B**) Levels of PDHA1 phosphorylation at Ser293 residue. (**C**) Levels of PDHA1 phosphorylation, normalized to the protein expression in the same samples. Data are presented as % from the indicated parameters in the control (Ctrl) animals. Significance of the treatment factors and differences between experimental groups are estimated and shown in the same way as in Figure 2.

**Figure 7 ijms-23-13186-f007:**
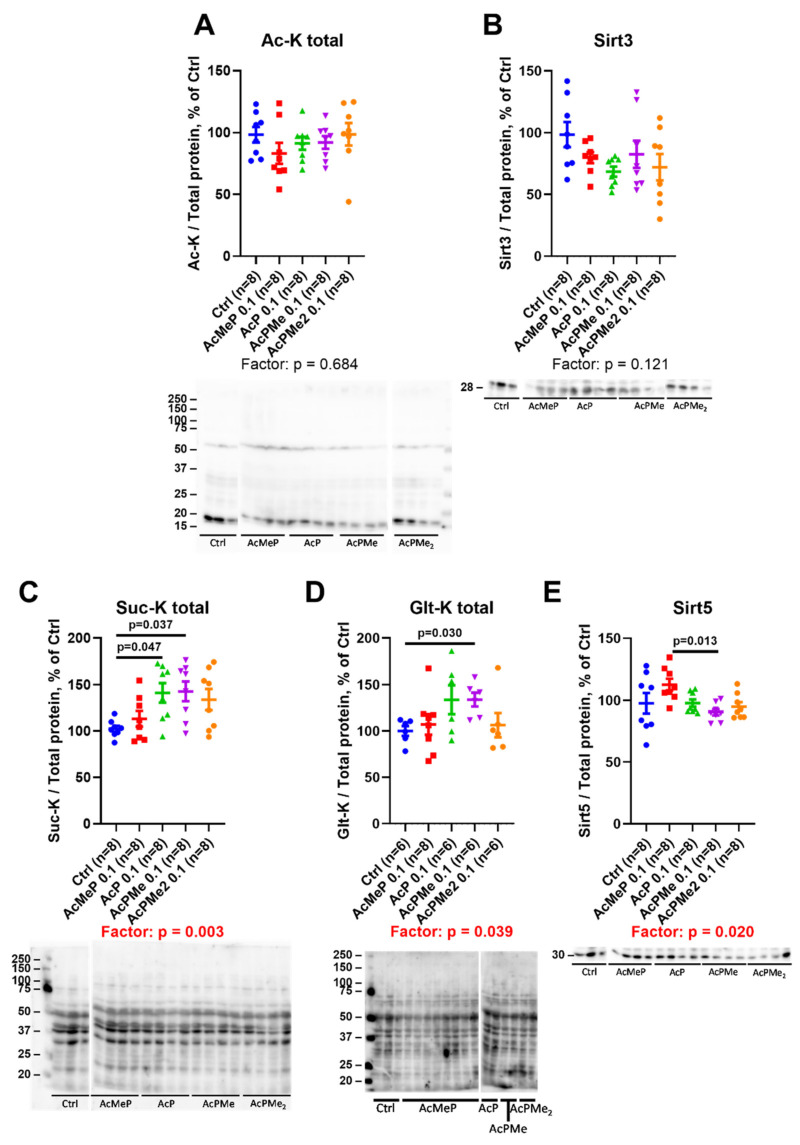
Changes in the components of acylation system in the cerebral cortex upon the administration of PDHC inhibitors. (**A**) Total protein acetylation (Ac-K). (**B**) Sirtuin 3 expression. (**C**) Total protein succinylation (Suc-K). (**D**) Total protein glutarylation (Glt-K). (**E**) Sirtuin 5 expression. Protein expression and acylation are determined by immunoblottings, shown below the graphs. Data are presented as % from the non-treated animals. Significant differences of the treatments and between experimental groups are estimated and shown in the same way as in Figure 2.

**Figure 8 ijms-23-13186-f008:**
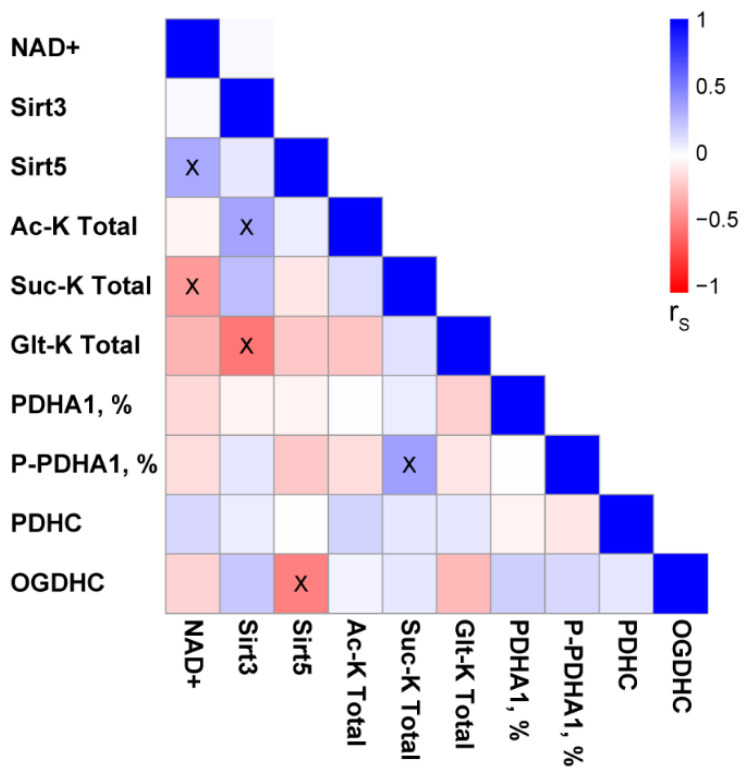
Pairwise correlations between the components of the brain acylation system. Spearman correlation coefficients (r_S_) and the significances of the correlations (crossed cells correspond to *p* < 0.05) are calculated for the indicated pairs of the parameters using the pooled sample of studied animals (*n* = 40).

## Data Availability

Raw data are available from the authors upon request.

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
