# Peer review of "Phosphonate Inhibitors of Pyruvate Dehydrogenase Perturb Homeostasis of Amino Acids and Protein Succinylation in the Brain"

_ijms, 2022, doi:10.3390/ijms232113186_

Round 1
Reviewer 1 Report
The paper “Phosphonate inhibitors of pyruvate dehydrogenase complex (PDHC) perturb homeostasis of amino acids and protein succinylation in the brain” by Artiukhov and coauthors is an interesting study addressing the role of interplay between activity of pyruvate dehydrogenase complex with amino acid metabolism. The major message of this study is that the inhibition of PDHC can be compensated by increased degradation of amino acids through the TCA cycle due to activation of oxoglutarate dehydrogenase complex (OGDHC). The manuscript is clearly written and conclusions are supported by experimental data.
There are several issues that should be discussed.
· Possible implication of the interplay between PDHC and OGDHC under hypoxic conditions should be addressed in the discussion section.
· It would be interesting to know how compensatory activation of OGDHC influence the rate of glycolysis.
· Is compensatory activation of OGDHC sufficient to maintain intracellular ATP levels?.
Author Response
We would like to thank the reviewer for the interesting questions to be addressed in the revised manuscript. Our answers are provided in the modified discussion as shown below.
- Possible implication of the interplay between PDHC and OGDHC under hypoxic conditions should be addressed in the discussion section.
We extended the discussion on the interplay, incorporating our published data on acute hypoxic stress as follows:
“…in the female rat cerebellum under physiological conditions, the levels of the metabolic indicators of the PDHC and OGDHC function, i.e. Ala and Glu, correspondingly, do not strongly correlate to each other or to the OGDHC activity. However, short-term exposures of these rats to hypoxia induce a strong positive correlation between the levels of Ala and Glu, and strong negative correlations of each indicator with the OGDHC activity (doi:10.3390/cells9010139; doi: 10.3389/fmed.2021.751639). Along with the changed correlations, simultaneous increases in the brain levels of alanine and glutamate after acute hypoxia indicate concerted changes in the substrate fluxes through the two multienzyme complexes, observed in the female rat cerebellum in response to the insult.”
- It would be interesting to know how compensatory activation of OGDHC influence the rate of glycolysis.
Our study indicates that activation of glycolysis cannot substitute for the impaired mitochondrial metabolism in the brain with inhibited PDHC. Moreover, decreased malate dehydrogenase activity suggests that the production of reducing equivalents in the cytosol is decreased. These considerations are now introduced into the discussion:
“By decreasing the acetyl-CoA entry to the TCA cycle, PDHC inhibition stimulates the entry of amino acids to the cycle through activation of the TCA-cycle-limiting OGDHC. The increased degradation of amino acids upon PDHC inhibition is manifested in increased activity of the ammonia-assimilating glutamine synthetase and decreased levels of glutamate and branched chain amino acids (Figure 2). Thus, the brain energy demands upon PDHC inhibition are addressed by usage of amino acids as alternative energy substrates. Obviously, activation of glycolysis cannot substitute for the impairment in energy production by the brain mitochondria with inhibited PDHC. Moreover, downregulation of malate dehydrogenase activity with increasing the PDHC inhibition manifests decreased entry of reducing equivalents from cytosol to mitochondria through the malate-aspartate shuttle, not excluding downregulation of the pyruvate production by glycolysis upon the PDHC inhibition. “
- Is compensatory activation of OGDHC sufficient to maintain intracellular ATP levels?
We extended the discussion to answer this question as follows:
It is worth noting that the metabolic and particularly energetic impact of the OGDHC upregulation, as well as the upregulation itself, are strongly conditional, depending on the physiological state, damage level, and cells/tissues under consideration (Graf et al., 2013; Araujo et al., 2012). Functional upregulation of the complex may support no decreases in the neuronal levels of NADP(H) or ATP up to a certain level of the OGDHC inhibition by its specific inhibitor succinyl phosphonate (Graf et al., 2013; Trofimova et al., 2012). However, administration of a high ethanol dose may also induce OGDHC upregulation, yet neuronal levels of both NAD(P)H and ATP significantly decrease in the medium with a high ethanol content (Graf et al., 2013). Furthermore, both ethanol and succinyl phosphonate may upregulate OGDHC, but their combination does not provide an additive effect, decreasing the OGDHC upregulation instead. Thus, the upregulation is impaired upon increasing damage, that may explain promotion of the ethanol-induced energetic impairment by succinyl phosphonate in cultured neurons (Graf et al., 2013). It should also be noted in this regard that an enzyme upregulation determined in homogenates under the standard assay conditions employing saturating concentrations of all the substrates and coenzymes, estimates the active enzyme quantity rather than the real substrate flux through the enzyme. The flux is determined not only by the level of the active enzyme, but also by the levels of its substrates/coenzymes. For instance, one of the mechanisms, effected in the brain to compensate for the decreased flux through the succinyl-phosphonate-inhibited OGDHC, is an increase in the brain levels of the OGDHC coenzyme ThDP (Mkrtchyan et al., 2016). Dependent on conditions, increases in the active enzyme quantity may thus occur by different means, not always detectable in the standard assay conditions. Overall, prediction of the metabolic impact of the changes in a single enzyme activity assayed under standard conditions, requires consideration of the enzymatic network and the levels of the system damage/reparation."
Reviewer 2 Report
1. Page 2, Line 71: Please move Figure 1 after where it first appears in the text.
2. Page 4, Line 140: Please state what RMSSD and ECG stand for.
3. Please have a conclusion section.
Author Response
We thank the reviewer for the suggestions which we gladly addressed to improve the manuscript.
- Page 2, Line 71: Please move Figure 1 after where it first appears in the text.
Figure 1 (lines 69-74) appears right after its first mentioning in the text (lines 66-68)
- Page 4, Line 140: Please state what RMSSD and ECG stand for.
Done
- Please have a conclusion section.
Done